# Ecological Model Explaining the Psychosocial Adaptation to COVID-19

**DOI:** 10.3390/ijerph19095159

**Published:** 2022-04-24

**Authors:** Tânia Gaspar, Teresa Paiva, Margarida Gaspar Matos

**Affiliations:** 1Sleep Medicine Center—CENC, 1070-068 Lisbon, Portugal; teresapaiva0@gmail.com; 2Centro Lusíada de Investigação em Serviço Social e Intervenção Social (CLISSIS), Universidade Lusíada, 1349-001 Lisboa, Portugal; 3Instituto de Saúde Ambiental (ISAMB), Faculdade de Medicina, Universidade de Lisboa, 1649-026 Lisbon, Portugal; 4Comprehensive Health Research Center (CHRC), Nova Medical School, Universidade Nova de Lisboa, 1169-056 Lisbon, Portugal; 5APPSYci, ISPA—University Institute, 1100-304 Lisbon, Portugal

**Keywords:** adjustment, mental health, health, attitudes and behavior, lifestyle, environmental health, COVID-19

## Abstract

The main objective of this study is to understand and characterize the adoption of an ecological perspective and the physical, psychological, social, and contextual health factors that may influence the adjustment to and mental health experiences during the COVID-19 pandemic. The study included 5479 participants, of which 3710 were female (67.7%), aged between 18 and 90 years old, with a mean age of 48.57 years (SD = 14.29), were considered three age groups: 21.5% up to 35 years old, 61.8% between 36 and 64 years old, and 16.7% 65 years old or more. The mental health and individual adjustment to the COVID-19 situation are explained by socio-demographic factors, health-related factors, lifestyles, attitudes and behaviors, lockdown experience, and place of residence. A better adaptation and mental health are observed among men, people with a higher educational level, people with lower sadness, nervousness, and burnout, and people whose health situation did not worsen with the pandemic. In terms of lifestyle, a better adaptation is related to a better quality of sleep, fewer nightmares, a higher practice of physical activity, and less consumption of processed foods and sweets. A better adaptation is also associated with lower levels of dependence on alcohol, TV, and SN (social networks) and a more positive experience of the lockdown imposed by the pandemic. Gender and age group differences in the described context were studied. Promoting a better adjustment and improved mental health when dealing with the COVID-19 requires an ecological understanding and multitarget interventions, targeting physical, mental, and social health together with the contextual environment.

## 1. Introduction

The pandemic had a huge impact on the lives of the populations. Adequate psychosocial adaptation to the COVID-19 pandemic should be understood from an ecological perspective.

The socio-economic environment contributes significantly to the health and health equity of both individuals and communities [1]. Pandemic management must consider individual and societal health and behaviors [2].

The pandemic caused by COVID-19 has challenged all countries worldwide and their governments and populations. The characteristics of different countries and regions and the pandemic management had an impact on morbidity and mortality [3,4].

Altogether it is mandatory to understand the factors linked to protection and those linked to risk while managing and adapting to the COVID-19 pandemic [5].

The exponential spread of the virus and the large-scale isolation and lockdown policies enacted by most governments were associated with the emergence of a wide range of psychological disorders, including panic, fear, anxiety, depression, and frustration [1]. The intra and inter-individual differences must be understood within a systemic perspective considering personal, interpersonal, and community factors [1,2].

At the intrapersonal level, gender, age, and health status had an influence on pandemic outcomes. Older people and men had higher mortality rates [6]. Women in the labor market were more severely affected than men [7]. Women are more likely to work from home, reduce working hours, and become unemployed [7,8,9]. Gender differences in stress management and mental health were also identified [10]: women showed more symptoms of depression, anxiety, and stress, and men showed more resilience to stress [11,12]. In females, both living alone or with more than six persons were associated with worse sleep quality and worse pandemic compliance, and a higher morbidity index [13,14].

Age also had an influence on how people adapted to the pandemic. In general, besides mortality, older people experienced a more negative impact in terms of worsening health status and revealing more comorbidities [15]. On the other hand, older people, women, and people with higher educational and socioeconomic levels more often adopted preventive behaviors in the face of COVID-19 (mask use, physical distance, social isolation, etc.). Working people and those with lower socioeconomic status may have found it more difficult to practice the recommended behaviors and be forced to engage in higher-risk behaviors [16,17].

Older people and people with chronic illness also report being more worried about the pandemic and afraid of catching the virus and its consequences [18,19] and reported increased levels of loneliness because of isolation and other COVID-19 restriction [20].

People with higher education show better stress management and more resilience in the face of the pandemic than people with lower education [12], unemployed people [21], people with lower education and lower health literacy report being less worried about becoming sick with COVID-19, many of them report that they do not believe they can become infected [19]. Good health literacy is associated with better knowledge and adequate protective behaviors [22].

Worry and fear are understandable emotional responses in stressful and uncertain contexts. People’s adherence to COVID-19 prevention measures is strongly affected by their knowledge and attitudes [23].

In the study by Wang and colleagues [24], the psychological impact of the lockdown was rated moderate to severe (16.5% reported moderate to severe depressive symptoms, 28.8% reported moderate to severe anxiety symptoms, and 8.1% reported moderate to severe stress levels).

People who considered that they were more satisfied with access to health information and those who maintained more specific precautionary measures (e.g., hand hygiene, mask wearing) showed fewer psychological impacts and lower levels of stress, anxiety, and depression. The long-term pandemic consequences and future economic and social issues are of major concern [25,26].

The pandemic crisis also brought changes in people’s lifestyles. In terms of sleep habits, many people have reported a lower quality of sleep [13,14,27]. Eating habits have also undergone changes associated with the COVID-19 pandemic. Overweight and obese people, highly educated people, and younger people often reported that they ate worse during confinement. There was an increase in the use of food delivery services compared to the time before confinement [28,29]. Regarding physical activity, there was a decrease in physical activity and an increase in sedentary behaviors during the pandemic and related lockdown periods in various populations, including children, adults, and patients with various health conditions [13,14,30,31]. Alcohol consumption has also changed with the COVID-19 pandemic. Increased alcohol consumption and abuse are often associated with depressive [26] and anxiety symptoms [32].

Social support, marriage, or cohabiting marital status may act as protective factors against the psychological distress, isolation, and social withdrawal caused by the pandemic [33]. In a study by Zhang and Ma [34], social support from family and friends emerged as a protective factor during a lockdown.

People’s personalities, attitudes, and behaviors contribute to understanding individual differences in pandemic compliance. People who are more pessimistic, have more negative attitudes, psychological inflexibility, and psychological problems showed worse management of the stress and uncertainty associated with the pandemic [35]. Dawson and Golijani-Moghaddam [36] also identified that psychological inflexibility and avoidance behavior are associated with distress and lower perceived well-being. Psychological flexibility emerges as a protective factor in adapting to and managing the acute and long-term COVID-19 challenges.

The level of knowledge and attitudes toward COVID-19 are different among groups with different ages, genders, education levels, marital statuses, and in different regions [37]. In a study by Yue et al. [38], it is concluded that higher education level, being female, being married, being a healthcare professional, and living in an urban area had a significant and positive impact on the knowledge about COVID-19.

Wu [39] concludes that social capital and social network interaction influence adaptation, behaviors, and attitudes toward COVID-19. The author mentions that this influence should be understood in a multidimensional and multilevel manner, considering its large complexity.

The studies conducted in the last two years in the area of COVID-19 [38,39,40] reveal that pandemic adaptation is critical in the short, medium, and probably long term in the adoption of preventive measures and related to biopsychosocial well-being.

In synthesis: the coronavirus pandemic (COVID-19) remains a universal health threat that tends to negatively affect people’s mental health and well-being [41,42]. It becomes critical to understand the different factors that influence the response to the pandemic at the individual level, at the level of health care and governance systems [5].

The purpose of this study is to understand and characterize from an ecological perspective the physical, psychological, social, and contextual health factors that influence psychosocial adaptation and mental health during the COVID-19 pandemic.

The adaptation to COVID-19 should be understood through the ecological model taking into account the influence at personal, interpersonal, and environmental levels. In view of this, we put forward the following research hypotheses:

**Hypothesis** **1.**
*Adaptation to COVID-19 is influenced by sociodemographic factors such as, gender, age, and Socio-Economic Status (SES).*


**Hypothesis** **2.**
*Adaptation to COVID-19 is influenced by participants’ physical and psychological health.*


**Hypothesis** **3.**
*Adaptation to COVID-19 is influenced by the participants’ lifestyle.*


**Hypothesis** **4.**
*Adaptation to COVID-19 is influenced by participants’ attitudes and behaviors towards COVID-19.*


**Hypothesis** **5.**
*Adaptation to COVID-19 is influenced by the subjective experience of lockdown.*


**Hypothesis** **6.**
*Adaptation to COVID-19 is influenced by factors in the environment, namely, the area of residence.*


## 2. Materials and Methods

### 2.1. Participants

Online surveys targeted several groups: (1) The general population; (2) Sleep Disorder patients (SDP); (3) professionals, COVID-19-involved: Medical doctors and nurses; (4) professionals, COVID-19-affected: Teachers, psychologists, and dentists [14].

This study includes 5479 participants, of which 3710 are female (67.7%) aged between 18 and 90 years old, with a mean age of 48.57 years (SD = 14.29). Age was organized into three age groups: 21.5% until 35 years old, 61.8% between 36 and 64 years old, and 16.7% with 65 or more years old, from all over the country.

### 2.2. Instrument

The total survey had 177 questions as follows: Demographics, health status; work; confinement characteristics, mood, attitudes, and behaviors; Calamity Experience Checklist; sleep; physical activity; multimedia use; nutrition; toxins and addictions.

The present paper used sociodemographic variables; symptoms evolution with COVID-19 (global health perception, insomnia, depression, anxiety, burnout, and fatigue); morbidities worse or better with COVID-19; lifestyles related variables: quality of sleep, nightmares, eating behavior, dependences, physical activity (PA), and work volume before and after the COVID-19 pandemic. Attitudes and behaviors during COVID-19 (negative and positive), experience during lockdown, the Calamity Experience Check List (CECL), which is the average of four Visual Analogical Scales (VAS) from 1 (low) to 10 (high), describing several mental states: depression, anxiety, irritability, and worries related to uncertainty. The Calamity Experience Checklist (CECL) demonstrated a good internal consistency value α = 0.85. The adjustment index of the Confirmatory Factor Analysis (CFA) was good, and the model shows good adequacy. Wald test and Lagrange Multiplier test (LM test) analysis did not reveal necessary changes to the model found. The standardized solution obtained in the confirmatory analysis model allowed us to verify that, in general, the items present good saturation, varying between β = 0.55 and β = 0.89. The confirmatory factor analysis revealed a robust and adequate model; the explained variance and the residual (disturbance) were appropriate and ranged from R2 = 0.31 e R2 = 0.79 [10,14] for housing/geography.

### 2.3. Procedure

The Survey Legend platform was used. Surveys were anonymous for adults (>18y), allowing data analysis and statistical use. The first page included: purpose, authors, ethical reference, contact person, and supporting entities. It was available online during the 1st COVID-19 wave, from April to August 2020.

The overall project was approved by CENC’s Ethical Committee on 1/2020. Consent was obtained from the participants. There was no funding, public or private, and no conflict of interests.

The questionnaire was administered online through Google Forms and was disseminated through social networks by the researchers and external partners of the study. The questionnaire was purpose-built by the research team and based on extensive previous experience and study in the areas under investigation. This paper is part of a larger study with a larger number of senior researchers and their research centers. The variables for this paper were selected, taking into account the objective of the study, the literature, and the available variables.

### 2.4. Data Analysis

The variables under study were descriptively separated into dichotomous variables and continuous variables (1 to 10).

Linear regression analyses were performed using the Calamity Experience Checklist as a dependent variable. Linear regression analyses were performed first by means of global analysis, secondly, separate analyses for men and women, and finally, analyses for the three age groups (up to 35 years old, between 36 and 64 years old, and 65 years old or older).

## 3. Results

In relation to the civil status of our participants, they were organized into two groups, 65.2% with a partner (married/union) and 34.8% without a partner (single/divorced/widow). Overall, 3687 participants (67.7%) had a bachelor’s/graduate degree or less, and 1758 participants (32.3%) had a master’s or Ph.D. degree.

Overall, 3250 participants (69.3%) are healthcare professionals, 813 participants (19.9%) are commerce, services, and industry professionals, 525 participants (11.2%) are education professionals, and 120 participants (2.6%) are science and technology-related professionals. Of these, 1077 (22.8%) participants reported sleep disorders.

Table 1 presents the descriptive variables (mean and standard deviation (SD).

The linear regression model presented in Table 2 has the CECL as a dependent variable and has an explanatory value of 46%. The model under study is robust F = 73.53 (32.2827), *p* < 0.001.

The calamity experience is best explained by gender and educational level. Health-related variables are best explained by depression, anxiety, and burnout and by the fact that the diseases have worsened with the COVID-19 pandemic. In relation to lifestyle, the calamity experience is best explained by the quality of sleep, nightmares, physical activity, consumption of processed food, sweets, and alcohol dependence. TV and new technologies dependency and work increased with the pandemic. At the level of attitudes and behaviors, the lower levels of positive attitudes and behaviors better explain the calamity experience. A more negative lockdown experience and the location of the house at the time of lockdown also contribute to explaining the calamity experience.

Table 3 presents the linear regression models with the calamity experience as the dependent variable for male and female participants. The linear regression model for female has an explanatory value of 41% and the model is robust F = 44.42, (31.1932), *p* < 0.001. For males, the explanatory value of the model is 50%, and the model is also robust F = 28.65, (31.863), *p* < 0.001. The model has a higher explanatory value for men than for women.

There are gender differences at the level of the variables that best explain the calamity experience. Age, marital status, and house location during lockdown contribute to explaining the calamity experience for men and not for women. On the other hand, educational level, sadness, physical activity, consumption of processed food and sweets, positive behaviors and stress, fear, and worries during lockdown contribute to explaining the calamity experience in women and not in men.

Table 4 presents the linear regression models with dependent variables for the calamity experience of different age groups. For younger participants (until 35 years old), the model under study is robust F = 16.24, (31.676), *p* < 0.001, and has an explanatory value of 41%. For participants aged between 36 and 64-years-old the model is also robust F = 50.69, (31.1877), *p* < 0.001, and the explanatory value of the model is 45%. For participants who are 65-years-old or older, the explanatory value of the model is 47%, and the model under study is also robust F = 7.62, (31.209), *p* < 0.001. The model has a higher explanatory value for older participants (36 years and older) than for younger participants (up to 35 years).

There are age differences at the level of the variables that explain the calamity experience: Gender, educational level, health status, physical activity, processed food, and candy consumption. TV dependence and positive behaviors contribute to explaining the calamity experience in participants aged 36 to 64 years and not for the other age groups.

All the following contribute to the explanation of the calamity experience in the younger participants group (up to 35 years and 36 to 64 years) and not for the older group (65 years or more): insomnia, burnout, nightmares, alcohol dependence, SN dependence, more work with the pandemic, positive attitudes, feeling loneliness, unexpected conflicts, stress/fear and worries during the lockdown period.

The morbidities worsened by the pandemic and the house location during the lockdown period have contributed to explaining the calamity experience in the older participant group (65 years or more) and not in the younger groups (group up to 35 years and group 36 to 64 years).

Finally, “can’t stand it” during the lockdown period explains the calamity experience in the older participants group (36 years old or more) and not in the younger group (until 35 years old), while sadness explains the calamity experience for the younger participants (until 35 years old) and not for the older participants (36 years old or more).

## 4. Discussion

The objective of this study is to understand and characterize the physical, psychological, social, and contextual health factors that influence the adaptation and the mental health in dealing with the COVID-19 pandemic from an ecological perspective.

The results allow us to conclude that the pandemic psychosocial adaptation, measured here through the management capacity and the levels of anxiety, depression, irritability, worry, and uncertainty, should be understood from an ecological perspective. In fact, the levels of adaptation and mental health are explained by sociodemographic variables, health-related factors, lifestyles, attitudes and behaviors, lockdown experience, and place of residence.

Better psychosocial adaptation and mental health were observed among men, people with a higher educational level, people with lower sadness, nervousness, and burnout, and those whose health situation did not worsen with the pandemic. In terms of lifestyles, we found better psychosocial adaptation among people who reported better sleep quality, fewer nightmares, more physical activity, and less consumption of processed foods and sweets. A better adaptation was also associated with lower levels of dependence on alcohol, TV, SNs, and a more positive experience during the lockdown imposed by the pandemic.

Understanding the level of adaptation and mental health in the face of COVID-19 needs to consider personal, social, and contextual factors, as all contribute to better or worse adaptation [1,3,5,10]. The present study allows us to identify which factors are involved and how the different factors have contributed to a better adaptation. At the sociodemographic level, gender and level of education contribute to explaining adaptation; in general, women and people with lower levels of education have a lower level of adaptation, which points to specificities for these groups and the need for specific prevention and intervention measures [7,8,12,43]. At the health level, we confirmed that participants reporting more health problems, as well as those whose health situation worsened during the pandemic, tend to report a lower level of psychosocial adaptation. These results allow us to conclude that people with greater health vulnerability from a biopsychological perspective are at greater risk of ineffectively managing the effects of the pandemic and, consequently, show greater difficulty in adapting and present with worse mental health indicators. Studies conducted by Shi et al. [23], focused on the relationship between fatigue and depressive symptoms. Wang et al. [24] and Stolz, Mayerl, and Freidl [20] studied the deepening psychological impact of the pandemic, lockdown, and consequent loneliness. Arslan et al. [35] argued that personality traits condition behaviors and attitudes and that, consequently, people who are more pessimistic and less open to change, show more difficulties in adapting to the challenges of the pandemic. Furthermore, a study by Wolf et al. [19] on chronically ill patients All of the above corroborates the results obtained in the present study. Volk et al. [40] developed a study to understand the influence of demographic and personality characteristics on the management of COVID-19. They conclude that both demographic and personality traits influence adaptation and the response to COVID-19. Demographic factors had fewer direct effects than personality traits.

Coping strategies targeting positive emotions are most effective in reducing psychological symptoms and in promoting mental health. Coping strategies associated with better mental health are positive reframing, acceptance, and humor; conversely, self-blame, venting, behavioral disengagement, and self-distraction were associated with poorer mental health [11,13,14].

The present results also have shown that lifestyles and health behaviors influence psychosocial adaptation and mental health during the pandemic and lockdown. People who reported more risk behaviors related to sleeping habits and quality [13,14,27], eating habits [28,29], physical activities [30,31], and addictions (TV, SN, and alcohol use) [32] tend to reveal less psychological wellbeing and greater difficulty in adapting to the challenges of the COVID-19 pandemic [26]. In relation to physical activity, the results of a study by Hargreaves et al. [31] showed that individuals with vigorous and moderate physical activity pre-COVID decreased intensity during the pandemic and lockdown. Conversely, moderately active individuals maintained or increased their physical activity practice.

Men, younger people, and single people are more vulnerable to developing alcohol-related risk behavior during a crisis and in stressful situations. In a study conducted by Grossman et al. [32], it was found that people reporting COVID-19-related stress consumed more alcoholic beverages in the pandemic than before, justifying this increase with the increased stress, greater availability/access to alcoholic beverages, and boredom.

By directly responding to the initial hypotheses, we have found that:

**Hypothesis** **1.**
*The adaptation to COVID-19 is influenced by sociodemographic factors, such as gender, age, and education level.*


The results show that there are differences related to age, gender, and level of education, in general, men, older people, and those with higher level of education show, in general, better levels of adaptation.

**Hypothesis** **2.**
*Adaptation to COVID-19 is influenced by the participants’ physical and psychological health.*


In relation to health-related factors, depression, anxiety, and burnout and the fact that the health conditions have worsened with the COVID pandemic negatively influence adaptation to COVID-19.

**Hypothesis** **3.**
*Adaptation to COVID-19 is influenced by participants’ lifestyle.*


In relation to lifestyle, the calamity experience was influenced by the quality of sleep, nightmares, physical activity, consumption of processed food, sweets, and alcohol dependence. TV and technologies dependence increased with the pandemic.

**Hypothesis** **4.**
*Adaptation to COVID-19 is influenced by attitudes and behaviors towards COVID-19.*


At the level of attitudes and behaviors lower of positive attitudes and behaviors are associated with worse adaptation to COVID-19.

**Hypothesis** **5.**
*Adaptation to COVID-19 is influenced by the subjective experience of lockdown.*


A more negative lockdown experience negatively influences adaptation to COVID-19.

**Hypothesis** **6.**
*Adaptation to COVID-19 is influenced by factors of the environment, namely the area of residence.*


Living in a rural area at the time of lockdown is negatively related to adaptation to COVID-19.

Figure 1 was created by the authors with the aim of illustrating the conceptual ecological model that was analysed, and results from this study.

The study of the model by gender reveals both similarities and differences for men and women. Regarding women, adaptation is better explained by lower sadness, physical activity, low consumption of processed foods and sweets, positive behaviors toward COVID-19, and better lockdown experience. Specifically, for men, adaptation is best explained by demographics such as (lower) age, (not alone) marital status, and (urban) home location.

The results allow an in-depth analysis of the factors that contribute most to gender differences in psychological adaptation to the pandemic. Similar to several other studies, it is confirmed that women, in general, show more difficulty in adapting at the level of biopsychological health, and the greater impact of sadness is highlighted; similar results found by Gurvich et al. [11] and Hou et al. [12] who study gender differences in depression and resilience. The challenges and uncertainty associated with the pandemic also affect people’s behaviors and attitudes [25]. In terms of eating behavior, women tend to have increased consumption of sweets and less healthy eating, Kandiah et al. [28] identify a similar pattern in which they conclude that women and overweight/obese people have worsened their eating habits during the pandemic. Women also reveal a less positive lockdown-related experience than men. In terms of work and reconciliation with family life, it was also women who became more frequently unemployed during the pandemic [21] and, on the other hand, are the ones who perform more risky jobs at the pandemic level, such as home care assistant, nursing assistant, cleaning, supermarket cashier, etc. [8,10,23].

Men seem to have been more affected by age, marital status, and house location. Modig et al. [6] also conclude that men and older people show higher morbidity and mortality outcomes which may affect their level of adaptation to the pandemic. In terms of marital status, men who are married/in union show higher levels of adaptation, social support, and marriage/cohabiting status can act as protective factors against maladaptation to the COVID-19 pandemic, unlike social isolation, which can negatively affect adaptation and mental health. Housing area also affects adaptation, with men living in rural areas or small towns showing worse adaptation. Yue et al. [37], in a study related to knowledge regarding COVID-19, concludes that women and people living in urban areas show more positive indicators.

The study of the model by age groups also allowed the identification of common and specific age factors. Within the youngest group (up to 35 years), adaptation is better explained by a better situation regarding insomnia, burnout, nightmares, alcohol and SN dependence, more work with the pandemic, positive attitudes, and lockdown experience. In the middle age group (36 to 64 years), adaptation is better explained by male gender, higher educational level, better health status, more physical activity, less consumption of processed foods and sweets, lower television dependence, and positive behaviors toward COVID-19. Among older participants (65 years and older), their adaptation is better explained by the existence of morbidities aggravated by the pandemic and by (non-urban) home location during the lockdown.

Age associated with health status, employment status, and social situation can affect pandemic adaptation. Younger people generally have better biopsychological health [6], their health is less affected, and they have fewer comorbidities [15,25]. Younger people encounter more positive life events, but on the other hand, they are the ones who encounter Non-COVID-19 stressful events [18]. In turn, possibly due to perceived risk and vulnerability [19], older people more readily adopt preventive behaviors in the face of COVID-19 (handwashing, distancing, isolation etc.) [16].

The results reinforce the complexity and multidimensionality of the mechanisms associated with psychosocial adaptation in the face of COVID-19. This aspect is mirrored in a study conducted over 36 countries [38], which concluded that it is critical to understand and characterize compliance to COVID-19 across different population groups and professional groups as it is a complex and multidimensional phenomenon. The study concludes that people’s psychosocial adaptation and well-being were related to their knowledge, attitude, and practices toward the COVID-19 pandemic.

The COVID-19 pandemic has and will have medium to long-term impacts on the lives and health of populations. It is important to use all innovative resources linked to new technologies, communication, and information mechanisms to combat and mitigate the negative effects of COVID-19 worldwide [44].

The obtained results discussed in this paper take into account the state-of-the-art methods and reinforce that our understanding of adaptation to COVID-19 should be studied from an ecological perspective [10,38,40].

## 5. Conclusions

It can be concluded from the present study that psychosocial adaptation to the COVID-19 pandemic is complex and multidimensional. To understand the factors that influence adaptation and mental health, it is essential to have an ecological perspective that integrates personal, social, and contextual factors.

At the individual level, gender, age, and education influence adaptation. Biopsychological health and worsening health status and lifestyles, particularly those related to sleeping habits, eating habits, physical exercise, and addictive behaviors, also interfere with the way people adapt to the pandemic and their mental health.

The existence of positive attitudes and behaviors toward COVID-19 shows a greater explanatory value for adaptation than the existence of negative attitudes and behaviors. Finally, lockdown experience and place of residence have also significantly impacted mental health and psychological well-being.

The study allows us to identify groups at higher risk; in general, women, older people, and those with lower levels of education have greater difficulty in adapting to the challenges posed by the pandemic.

The most relevant limitations are the diversity and breadth of the participants. The sample size is large and includes several subgroups, which enriches the study and allows for future papers on the in-depth study of different populations, such as health professionals, and participants with sleep disorders, among others. Another reflection concerns the biopsychosocial health situation of the participants in the pre-pandemic moment. In the questions asked, we asked the participants to evaluate the situation during the pandemic compared to the pre-pandemic period to assess whether or not there was a worsening in health. The answer implies the perception of each participant and not an objective evaluation.

The results are considered to provide a comprehensive view of the complexity of the factors and interconnection of factors that arise in relation to better or worse adaptation to the pandemic COVID-19. The results obtained are relevant information for mental health prevention and adaptation to the COVID-19 pandemic and can be considered for other global stressful events and changes.

The whole population would benefit from multidisciplinary and multisectoral intervention, aiming at promoting well-being and adaptation to the new reality. The groups most at risk need more focused and intensive interventions on the identified needs, perceptions of sadness and nervousness, healthy lifestyle, and prioritizing the promotion of positive behaviors and attitudes towards protection from COVID-19 and protection from other emerging health, social and environmental crises must be stressed.

Key Points:Psychosocial adaptation to COVID-19 should be understood within an ecological perspective, influenced by individual, social and contextual factors;Psychosocial adaptation to COVID-19 is influenced by sociodemographic factors, health and lifestyles, behaviors and attitudes face COVID-19, lockdown experience, and housing conditions/location;In general, women, older people, and those with lower levels of education have greater difficulty in adapting to the challenges posed by the pandemic;The results obtained are relevant information for mental health prevention and adjustment to the COVID-19 pandemic and can be considered for other global stressful events and changes;The whole population would benefit from multidisciplinary and multisectoral intervention aiming at promoting well-being and adaptation to the new reality.

## Figures and Tables

**Figure 1 ijerph-19-05159-f001:**
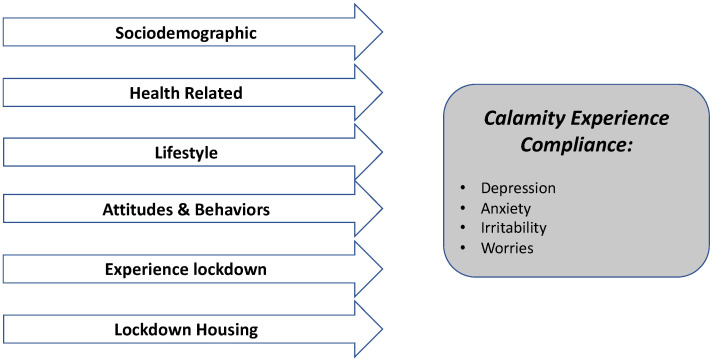
Ecological conceptual model resulting from the study.

**Table 1 ijerph-19-05159-t001:** Descriptive variables (range, mean, and standard deviation (SD).

	Range	Mean	SD
Calamity Experience Checklist *	1 (less) to 10 (high)	4.80	2.05
Healthy	0 (no) and 1 (yes)	0.29	0.45
Insomnia	0 (no) and 1 (yes)	0.19	0.39
Sadness	0 (no) and 1 (yes)	0.0	0.29
Nervousness	0 (no) and 1 (yes)	0.12	0.33
Burnout	0 (no) and 1 (yes)	0.11	0.32
Fatigue	0 (no) and 1 (yes)	0.07	0.26
Morbidities Worse with COVID-19	0 (no) and 1 (yes)	0.26	0.44
Morbidities Better with COVID-19	0 (no) and 1 (yes)	0.38	0.49
Sleep Quality	1 (less) to 10 (high)	5.70	2.18
Nightmares	0 (no) and 1 (yes)	0.26	0.44
Physical Activity	0 (no) and 1 (yes)	0.28	0.45
Fruit & vegetables	0 (no) and 1 (yes)	0.67	0.47
Processed Food	0 (no) and 1 (yes)	0.74	0.44
Sweets	0 (no) and 1 (yes)	0.87	0.34
Alcohol Dependence	1 (less) to 10 (high)	1.47	1.16
TV Dependence	1 (less) to 10 (high)	3.33	2.08
SN Dependence	1 (less) to 10 (high)	3.64	2.38
Work intensity with COVID-19	0 (less or equal) and 1 (higher)	0.48	0.50
Negative attitudes	0 (no) and 1 (yes)	0.11	0.31
Positive attitudes	0 (no) and 1 (yes)	0.10	0.30
Negative behaviors	0 (no) and 1 (yes)	0.14	0.35
Positive Behaviors	0 (no) and 1 (yes)	0.16	0.37
Lockdown—fed up/tired	0 (no) and 1 (yes)	0.42	0.49
Lockdown—cannot stand	0 (no) and 1 (yes)	0.06	0.24
Lockdown—loneliness	0 (no) and 1 (yes)	0.12	0.33
Lockdown—unexpected conflicts	0 (no) and 1 (yes)	0.08	0.28
Lockdown—stress/fears/worries	0 (no) and 1 (yes)	0.03	0.18
Lockdown—Location House	1 (City) and 2 (Rural/village)	1.26	0.44

* (CECL) (How_is_your_depression + How_is_your_anxiety + How_is_your_irritability + How_are_your_worries)/4.

**Table 2 ijerph-19-05159-t002:** Linear regression—dependent variable Calamity Experience Checklist.

		Unstandardised Coefficients		Standardised Coefficients	
		B	*SE*	β	t
	(Constant)	3.616	0.364		9.935 ***
Socio demographic	Age	−0.005	0.003	−0.031	−1.767 (n.s.)
	Gender	0.269	0.066	0.061	4.052 ***
	Education Level	−0.120	0.063	−0.028	−1.900 *
	Marital status	0.111	0.064	0.026	1.748 (n.s.)
Health related	Healthy	−0.239	0.182	−0.055	−1.311 (n.s.)
	Insomnia	−0.064	0.082	−0.012	−0.782 (n.s.)
	Sadness	0.617	0.113	0.082	5.452 ***
	Nervousness	0.411	0.098	0.065	4.187 ***
	Burnout	0.701	0.095	0.115	7.357 ***
	Fatigue	0.179	0.114	0.023	1.562 (n.s.)
	Morbidities Worse with COVID-19	0.531	0.069	0.116	7.679 ***
	Morbidities Better with COVID-19	0.069	0.072	0.014	0.968 (n.s.)
Lifestyle	Sleep Quality	−0.210	0.015	−0.224	−13.757 ***
	Nightmares	0.456	0.070	0.100	6.492 ***
	Physical Activity	−0.158	0.062	−0.036	−2.523 **
	Fruit & vegetables	0.110	0.062	0.025	1.781 (n.s.)
	Processed Food	0.189	0.073	0.038	2.576 **
	Sweets	0.236	0.090	0.037	2.625 **
	Alcohol Dependence	0.147	0.026	0.082	5.710 ***
	TV Dependence	0.067	0.016	0.067	4.296 ***
	SN Dependence	0.090	0.014	0.104	6.274 ***
	Work intensity with COVID-19	0.270	0.060	0.066	4.529 ***
Attitudes and behaviors	Negative attitudes	0.079	0.122	0.013	0.643 (n.s.)
	Positive attitudes	−0.499	0.097	−0.076	−5.115 ***
	Negative behaviors	0.060	0.088	0.010	0.684 (n.s.)
	Positive Behaviors	−0.254	0.078	−0.047	−3.247 ***
Experience lockdown	Lockdown—fed up/tired	0.551	0.063	0.134	8.773 ***
	Lockdown—cannot stand	0.513	0.135	0.061	3.807 ***
	Lockdown—loneliness	0.465	0.097	0.077	4.804 ***
	Lockdown—unexpected conflicts	0.656	0.109	0.096	5.991 ***
	Lockdown—stress/fears/worries	0.620	0.160	0.055	3.865 ***
Housing/ geography	Lockdown—Location House	0.182	0.066	0.038	2.752 **

Dependent variable—Calamity Experience Checklist. Note: *** *p* < 0.001; ** *p* < 0.01; * *p* < 0.05. (n.s.) = non-statistically significant differences.

**Table 3 ijerph-19-05159-t003:** Linear regression—variable dependent Calamity Experience Checklist—gender differences.

	Female				Male			
	Unstandardised Coefficients		Standardised Coefficients		Unstandardised Coefficients		Standardised Coefficients	
	B	*SE*	β	t	B	*SE*	β	t
(Constant)	4.251	0.407		10.457 ***	3.619	0.601		6.027 ***
Age	−0.002	0.004	−0.013	−0.592 (n.s.)	−0.009	0.004	−0.060	−2.060 *
Education Level	−0.168	0.076	−0.041	−2.192 *	−0.028	0.114	−0.006	−0.247 (n.s.)
Marital status	0.054	0.075	0.013	0.720 (n.s.)	0.288	0.123	0.062	2.336 *
Healthy	−0.274	0.218	−0.065	−1.259 (n.s.)	−0.350	0.335	−0.079	−1.044 (n.s.)
Insomnia	−0.043	0.097	−0.009	−0.440 (n.s.)	−0.070	0.162	−0.012	−0.432 (n.s.)
Sadness	0.627	0.130	0.091	4.828 ***	0.414	0.238	0.046	1.738 (n.s.)
Nervousness	0.384	0.111	0.067	3.466 ***	0.535	0.224	0.065	2.388 *
Burnout	0.594	0.114	0.102	5.202 ***	0.900	0.176	0.137	5.115 ***
Fatigue	0.148	0.129	0.021	1.150 (n.s.)	0.373	0.262	0.036	1.422 (n.s.)
Morbidities Worse with COVID-19	0.484	0.082	0.111	5.875 ***	0.699	0.129	0.139	5.427 ***
Morbidities Better with COVID-19	0.050	0.090	0.010	0.556 (n.s.)	0.107	0.119	0.022	0.898(n.s.)
Sleep Quality	−0.188	0.019	−0.206	−10.107 ***	−0.234	0.028	−0.240	−8.495 ***
Nightmares	0.460	0.082	0.108	5.643 ***	0.546	0.143	0.098	3.818 ***
Physical Activity	−0.175	0.079	−0.039	−2.211 *	−0.129	0.102	−0.031	−1.264 (n.s.)
Fruit & vegetables	0.123	0.073	0.030	1.677 (n.s.)	0.127	0.118	0.026	1.075 (n.s.)
Processed Food	0.235	0.089	0.049	2.629 **	0.092	0.129	0.018	0.715 (n.s.)
Sweets	0.328	0.112	0.052	2.919 **	0.033	0.151	0.005	0.218 (n.s.)
Alcohol Dependence	0.129	0.034	0.067	3.778 ***	0.163	0.039	0.103	4.140 ***
TV Dependence	0.044	0.019	0.047	2.374 *	0.109	0.029	0.103	3.779 ***
SN Dependence	0.095	0.017	0.115	5.508 ***	0.089	0.026	0.097	3.370 ***
Work intensity with COVID-19	0.232	0.072	0.058	3.207 ***	0.313	0.107	0.074	2.925 **
Negative attitudes	0.197	0.141	0.035	1.393 (n.s.)	−0.197	0.253	−0.026	−0.779 (n.s.)
Positive attitudes	−0.647	0.126	−0.097	−5.151 ***	−0.279	0.155	−0.047	−1.802 (n.s.)
Negative behaviors	0.083	0.105	0.015	0.789 (n.s.)	−0.023	0.162	−0.004	−0.144 (n.s.)
Positive Behaviors	−0.336	0.097	−0.063	−3.481 ***	−0.109	0.135	−0.020	−0.806 (n.s.)
Lockdown—fed up/tired	0.496	0.077	0.124	6.468 ***	0.616	0.111	0.149	5.575 ***
Lockdown—cannot stand	0.555	0.156	0.072	3.564 ***	0.490	0.276	0.048	1.771 (n.s.)
Lockdown—loneliness	0.333	0.112	0.060	2.968 **	0.864	0.196	0.124	4.415 ***
Lockdown—unexpected conflicts	0.555	0.129	0.086	4.310 ***	0.841	0.212	0.110	3.955 ***
Lockdown—stress/fears/worries	0.718	0.178	0.071	4.038 ***	0.093	0.393	0.006	0.236 (n.s.)
Lockdown—Location House	0.122	0.081	0.027	1.509 (n.s.)	0.306	0.116	0.064	2.635 **

Dependent variable—Calamity Experience Checklist. Note. *** *p* < 0.001; ** *p* < 0.01; * *p* < 0.05; (n.s.) = non-statistically significant differences.

**Table 4 ijerph-19-05159-t004:** Linear regression—variable dependent Calamity Experience Checklist—age groups differences.

	Until 35 Years				Between 36 and 64 Years				65 Years or More			
	Unstandardised Coefficients		Standardised Coefficients		Unstandardised Coefficients		Standardised Coefficients		Unstandardised Coefficients		Standardised Coefficients	
	B	*SE*	β	t	B	*SE*	β	t	B	*SE*	β	t
(Constante)	3.869	0.678		5.706	3.116	0.389		8.015 ***	1.017	1.672		0.608 (n.s.)
Gender	0.253	0.145	0.053	1.739 (n.s.)	0.237	0.081	0.054	2.940 **	0.361	0.223	0.095	1.617 (n.s.)
Education Level	0.154	0.133	0.035	1.150 (n.s.)	−0.159	0.080	−0.034	−1.991 *	−0.412	0.252	−0.085	−1.637 (n.s.)
Marital status	0.158	0.123	0.039	1.290 (n.s.)	0.100	0.082	0.022	1.228 (n.s.)	−0.171	0.234	−0.040	−0.731 (n.s.)
Healthy	0.461	0.252	0.058	1.832 (n.s.)	0.632	0.133	0.087	4.742 ***	0.052	0.450	0.007	0.116 (n.s.)
Insomnia	0.313	0.142	0.072	2.205 *	0.615	0.085	0.134	7.275 ***	0.468	0.258	0.098	1.810 (n.s.)
Sadness	−0.876	0.332	−0.220	−2.637 **	0.031	0.224	0.007	0.138 (n.s.)	2.380	1.496	0.542	1.590 (n.s.)
Nervousness	−0.029	0.183	−0.005	−0.160 (n.s.)	−0.122	0.098	−0.024	−1.249 (n.s.)	0.022	0.316	0.004	0.069 (n.s.)
Burnout	0.381	0.187	0.069	2.033 *	0.508	0.123	0.079	4.132 ***	0.367	0.429	0.048	0.856 (n.s.)
Fatigue	0.639	0.198	0.107	3.233 ***	0.682	0.115	0.114	5.934 ***	1.081	0.412	0.149	2.624 **
Morbidities Worse with COVID−19	0.153	0.283	0.017	0.540 (n.s.)	0.112	0.132	0.016	0.847 (n.s.)	0.989	0.481	0.116	2.058 *
Morbidities Better with COVID-19	0.191	0.170	0.034	1.123 (n.s.)	0.051	0.085	0.010	0.599 (n.s.)	0.005	0.232	0.001	0.021 (n.s.)
Sleep Quality	−0.277	0.032	−0.297	−8.590 ***	−0.182	0.019	−0.195	−9.693 ***	−0.210	0.051	−0.237	−4.112 ***
Nightmares	0.278	0.131	0.069	2.129 *	0.578	0.089	0.121	6.493 ***	0.513	0.288	0.099	1.785 (n.s.)
Physical Activity	0.015	0.124	0.004	0.119 (n.s.)	−0.192	0.079	−0.042	−2.420 *	−0.271	0.196	−0.071	−1.382 (n.s.)
Fruit & vegetables	0.187	0.128	0.044	1.458 (n.s.)	0.046	0.076	0.011	0.607 (n.s.)	0.372	0.206	0.094	1.806 (n.s.)
Processed Food	0.283	0.200	0.043	1.415 (n.s.)	0.188	0.085	0.039	2.225 *	−0.001	0.196	0.000	−0.003 (n.s.)
Sweets	0.312	0.194	0.050	1.611 (n.s.)	0.299	0.112	0.047	2.677 **	−0.343	0.256	−0.067	−1.337 (n.s.)
Alcohol Dependence	0.148	0.055	0.082	2.682 **	0.142	0.032	0.078	4.454 ***	0.117	0.083	0.078	1.408 (n.s.)
TV Dependence	0.035	0.029	0.040	1.224 (n.s.)	0.082	0.020	0.081	4.199 ***	0.078	0.062	0.075	1.250 (n.s.)
SN Dependence	0.097	0.027	0.119	3.624 ***	0.089	0.018	0.098	5.049 ***	0.073	0.064	0.070	1.145 (n.s.)
Work intensity with COVID-19	0.300	0.121	0.076	2.477 **	0.261	0.073	0.064	3.592 ***	−0.058	0.249	−0.012	−0.232 (n.s.)
Negative attitudes	0.015	0.237	0.003	0.063 (n.s.)	0.186	0.150	0.029	1.235 (n.s.)	−0.590	0.606	−0.060	−0.974 (n.s.)
Positive attitudes	−0.821	0.219	−0.115	−3.741 ***	−0.456	0.118	−0.072	−3.864 ***	−0.011	0.298	−0.002	−0.037 (n.s.)
Negative behaviors	−0.001	0.146	0.000	−0.004 (n.s.)	0.147	0.118	0.022	1.244 (n.s.)	0.300	0.362	0.047	0.829 (n.s.)
Positive Behaviors	−0.160	0.161	−0.030	−0.996 (n.s.)	−0.345	0.098	−0.062	−3.508 ***	0.093	0.234	0.021	0.397 (n.s.)
Lockdown—fed up/tired	0.473	0.129	0.120	3.663 ***	0.552	0.077	0.134	7.193 ***	0.739	0.211	0.189	3.504 ***
Lockdown—cannot stand	0.339	0.244	0.050	1.390 (n.s.)	0.610	0.173	0.067	3.524 ***	1.227	0.552	0.125	2.223 *
Lockdown—loneliness	0.528	0.171	0.106	3.094 **	0.419	0.125	0.066	3.346 ***	0.530	0.395	0.073	1.343 (n.s.)
Lockdown—unexpected conflicts	0.619	0.206	0.104	3.006 **	0.604	0.134	0.088	4.494 ***	1.457	0.644	0.122	2.263 (n.s.)
Lockdown—stress/fears/worries	0.940	0.375	0.075	2.508 *	0.608	0.185	0.057	3.284 ***	−0.052	0.777	−0.004	−0.067 (n.s.)
Lockdown—Location House	0.243	0.147	0.049	1.653 (n.s.)	0.116	0.080	0.025	1.447 (n.s.)	0.404	0.208	0.100	1.941 *

Dependent variable—Calamity Experience Checklist. Note. *** *p* < 0.001; ** *p* < 0.01; * *p* < 0.05. (n.s.) = non-statistically significant differences.

## Data Availability

Not applicable.

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
