# Peer review of "Ecological Model Explaining the Psychosocial Adaptation to COVID-19"

_ijerph, 2022, doi:10.3390/ijerph19095159_

Round 1

Reviewer 1 Report

This is a well-designed project with appropriate age distribution, civil status. education, lifestyle, and geography. The results are reliable, and the conclusions help people to understand the complicated influence of the COVID-19 pandemic. I have two minor suggestions.

  1. When you consider the psychological factors, did you consider the pre-existing mental issues? That may not be directly caused by the pandemic.
  2. In addition to the marriage status, the family size could also be a factor that affects the adaption of COVID-19.

Author Response

Dear Reviewer
Thank you very much for the comments and suggestions that contribute to the improvement of our work, following the changes made. Any clarification or change needed just let us know  Best regards Tania

  1. When you consider the psychological factors, did you consider the pre-existing mental issues? That may not be directly caused by the pandemic.

This question is very relevant and we will include it in the limitations. The questions asked the person to compare him/herself to the pre-covid period, has the limitations of being an auto-answer but shows us the perception of worsening or not according to the participant.

  1. In addition to the marriage status, the family size could also be a factor that affects the adaption of COVID-19.

Estimated revisor marriage status was included in the linear regression analyses (table 2, 3 and 4), we found that it has no significant explanatory effect for the overall model nor for the different age groups but there is a statistically significant explanatory value for men (table 3)

Reviewer 2 Report

Gaspar and colleagues attempted to identify factors association with adaptation to the COVID-19 pandemic via a large-scale online survey. They have a large sample size of 5,479 participants, this is very useful especially considering that a long list of factors on health, societal, economic and psychological status were collected via a comprehensive questionnaire with 177 questions. While this rich dataset is encouraging, the data analysis and reporting themselves were unfortunately underdeveloped and require substantial additional work before further consideration. In particular, the only statistical test being used, linear regression, appears to be not appropriate for the complexity of the data involved. This is a major flaw of the study and a more careful analysis plan is needed. Other suggestions for the authors to consider -

  1. The title describes about ecological factors influencing compliance with COVID-19. This is ambiguous as one may misinterpret the work as primarily about compliance with public health and social measures (PHSMs) for COVID-19. Instead, the work is about adaptation of adults during the pandemics. The title thus needs to be rephrased to better reflect the main theme of the work. In addition, the term “compliance with COVID-19” should be replaced with a clearer term (e.g., psychological adaptation to pandemic) throughout the work to avoid confusion.

  1. Abstract: It is stated that “In terms of lifestyle, a better situation is found in people who report better quality of sleep, fewer nightmares, higher practice of physical activity and less consumption of processed foods and sweets.”. The authors may have conflated contributors and outcomes. While more physical exercise can contribute to a better mental wellness, it is not clear how better quality of sleep and fewer nightmares can do so as these are not generally controllable and are outcomes rather than contributors. A better justification and discrimination of contributing factors and outcome measures are needed in data analysis, result reporting and finding interpretation.

  1. Introduction: (a) The introduction is overly long and some sections appear to be better fit for Discussion which appears to be underdeveloped in terms of word counts compared with Introduction. (b) Moreover, the flow is a bit fragmented and some paragraphs can be combined to maintain coherence when reading through the work. E.g., “Older people and people with chronic illness also…” and “Older adults reported increased levels of loneliness..” can be consolidated into a single paragraph that focuses on discussing situation in older adults.

  1. The Method part should focus on describing methodology and not be mixed with results (e.g, descriptions of demographic characteristics of online survey participants on age distribution, gender ratio and education level). These data should be reported in Results.

  1. The online questionnaire with 177 questions should be provided as a supplementary material and a translated version into English is preferred for wider circulation where appropriate.

  1. The statistical part in Method should be substantially reinformed with more technical details. For example, one may assume that multivariate analysis have been performed given that an array of demographic, socio-economical, societal and health-related factors were analysed in the study.

  1. Results: The proportion of participants in the four main groups, namely, general, those with pre-existing sleeping disorder, COVID-19-related professionals and COVID-19 affected professionals should be provided and incorporated into data analysis.

  1. Linear regression was performed among the three age groups (young adults, middle-aged adults and older adults and multiple contributors were shortlisted. How were these factors controlled for collinearity and confounding factors?

  1. The manuscript would benefit from professional English-editing.

Author Response

Dear Reviewer
Thank you very much for the comments and suggestions that contribute to the improvement of our work, following the changes made. any clarification or change needed just let us know Best regards Tania

Gaspar and colleagues attempted to identify factors association with adaptation to the COVID-19 pandemic via a large-scale online survey. They have a large sample size of 5,479 participants, this is very useful especially considering that a long list of factors on health, societal, economic and psychological status were collected via a comprehensive questionnaire with 177 questions. While this rich dataset is encouraging, the data analysis and reporting themselves were unfortunately underdeveloped and require substantial additional work before further consideration. In particular, the only statistical test being used, linear regression, appears to be not appropriate for the complexity of the data involved. This is a major flaw of the study and a more careful analysis plan is needed.

Dear reviewer, thank you for your comment, exactly because of the complexity of the study, the diversity and type of variables used to perform a more advanced analysis as a structural equations model would not be able to include all the variables under study in the model. Therefore, we have chosen to perform the regression analysis including all the variables under study and we are working on an article in which we will use the structural equations model, but only with a reduced number of variables suitable for the analysis. However, we consider that our option has its advantages because it allows us, from a more global and integrated perspective, to understand and characterise the influence of numerous factors on the adaptation to the covid-19 pandemic, from a bio-psycho-social and environmental perspective.

Other suggestions for the authors to consider -

  1. The title describes about ecological factors influencing compliance with COVID-19. This is ambiguous as one may misinterpret the work as primarily about compliance with public health and social measures (PHSMs) for COVID-19. Instead, the work is about adaptation of adults during the pandemics. The title thus needs to be rephrased to better reflect the main theme of the work. In addition, the term “compliance with COVID-19” should be replaced with a clearer term (e.g., psychological adaptation to pandemic) throughout the work to avoid confusion.

we have changed the term "compliance" to "psychosocial adaptation" in the title and document

  1. Abstract: It is stated that “In terms of lifestyle, a better situation is found in people who report better quality of sleep, fewer nightmares, higher practice of physical activity and less consumption of processed foods and sweets.”. The authors may have conflated contributors and outcomes. While more physical exercise can contribute to a better mental wellness, it is not clear how better quality of sleep and fewer nightmares can do so as these are not generally controllable and are outcomes rather than contributors. A better justification and discrimination of contributing factors and outcome measures are needed in data analysis, result reporting and finding interpretation.

we changed the wording, here we do not stipulate causality, only relationship, we verified a relationship between better adaptation to the pandemic and better sleep quality and fewer nightmares, nightmares can be considered outcomes, sleep quality can already be considered a bidirectional relationship ...

  1. Introduction: (a) The introduction is overly long and some sections appear to be better fit for Discussion which appears to be underdeveloped in terms of word counts compared with Introduction. (b) Moreover, the flow is a bit fragmented and some paragraphs can be combined to maintain coherence when reading through the work. E.g., “Older people and people with chronic illness also…” and “Older adults reported increased levels of loneliness..” can be consolidated into a single paragraph that focuses on discussing situation in older adults.

The introduction was revised, reorganised and shortened and the discussion was strengthened taking into account the reviewer's recommendations.

  1. The Method part should focus on describing methodology and not be mixed with results (e.g, descriptions of demographic characteristics of online survey participants on age distribution, gender ratio and education level). These data should be reported in Results.

Dear reviewer it is generally accepted and recommended that the description of the participants refer information about the descriptives regarding gender, age, if you agree I will keep it that way. I will move the education level information and the information from table 1 to the results

The online questionnaire with 177 questions should be provided as a supplementary material and a translated version into English is preferred for wider circulation where appropriate.

The team prefers to keep the instrument without disclosure since it is in the validation phase. Is this mandatory?

  1. The statistical part in Method should be substantially reinformed with more technical details. For example, one may assume that multivariate analysis have been performed given that an array of demographic, socio-economical, societal and health-related factors were analysed in the study.

Dear reviewer, thank you for your comment, exactly because of the complexity of the study, the diversity and type of variables used to perform a more advanced analysis as a structural equations model would not be able to include all the variables under study in the model. Therefore, we have chosen to perform the regression analysis including all the variables under study and we are working on an article in which we will use the structural equations model, but only with a reduced number of variables suitable for the analysis. However, we consider that our option has its advantages because it allows us, from a more global and integrated perspective, to understand and characterise the influence of numerous factors on the adaptation to the covid-19 pandemic, from a bio-psycho-social and environmental perspective.

  1. Results: The proportion of participants in the four main groups, namely, general, those with pre-existing sleeping disorder, COVID-19-related professionals and COVID-19 affected professionals should be provided and incorporated into data analysis.

The information on the frequency of participants in the different groups was integrated in the results. This study aims to make a global study of the whole population involved from an ecological perspective, not to study specific populations. Future publications are being prepared with in-depth studies of each of the groups, namely health professionals and people with sleep disorders and with a more restricted number of variables.

  1. Linear regression was performed among the three age groups (young adults, middle-aged adults and older adults and multiple contributors were shortlisted. How were these factors controlled for collinearity and confounding factors?

Table 4 reveals that the model maintains the behaviour for the different age groups and the focus of the paper is not to study age, so we chose not to delve deeper.

The manuscript would benefit from professional English-editing.

Done

Reviewer 3 Report

Major comments:

The study and its results might be of interest for readers but the writing of this article needs improvement.

First of all, there is a clear difference in how discussion and conclusions were written (they are better) compared to the rest of the article.

English has to be improved along the entire article but mainly in introduction, materials and methods and results.

There is a lack of consistency in paying attention to details (e.g. COVID-19 is written in so many ways across the lines).

Introduction is far too long. In fact is more like a mini-review. Most of the information from introduction has to be moved to discussion.

Materials and methods: how was the questionnaire administered and how were the subjects selected? What are the databases used? What was the respond rate? What questionnaire was used? Is it an original one? Than how were the questions (items) selected? Is there a reference for that?

Information in table 1 is somehow redundant with table 2. They might be combined or, table 1 mentioned in supplement material only. The administered questionnaire might be added to supplemental material as well.

Minor comments:

page 2 line 48: this affirmation require a reference.

page 2 line 66: A study by... whom?

page 2 line 92: In the study by ... whom?

page 4 line 157-159: unclear. Needs rephrase.

page 4 line 187: SES - needs explanation first time appear in the text

page 4 lines 195-196: unclear. Please rephrase.

page 4 line 202: Paiva et al 2021b. Reference number?

page 5 line 223: CFA. Explanation?

page 5 line 227-229: unclear. Please rephrase and provide details, explanations. Tome et al needs reference number.

page 7 Table 3: what is n.s.? Explanation at the bottom of table.

page 11 lines: 285-287: unclear. please rephrase.

page 13 lines 374-375: unclear. please rephrase.

Author Response

Dear Reviewer
Thank you very much for the comments and suggestions that contribute to the improvement of our work, following the changes made. any clarification or change needed just let us know Best regards Tania

The study and its results might be of interest for readers but the writing of this article needs improvement.

First of all, there is a clear difference in how discussion and conclusions were written (they are better) compared to the rest of the article.

English has to be improved along the entire article but mainly in introduction, materials and methods and results.

The introduction was revised, reorganised and shortened and the discussion was strengthened taking into account the reviewer's recommendations.

There is a lack of consistency in paying attention to details (e.g. COVID-19 is written in so many ways across the lines).

The article was revised and the writing and terms homogenised

Introduction is far too long. In fact is more like a mini-review. Most of the information from introduction has to be moved to discussion.

The introduction was revised, reorganised and shortened and the discussion was strengthened taking into account the reviewer's recommendations.

Materials and methods: how was the questionnaire administered and how were the subjects selected? What are the databases used? What was the respond rate? What questionnaire was used? Is it an original one? Than how were the questions (items) selected? Is there a reference for that?

The questionnaire was administered online through the Google forms platform and was disseminated through the social networks of the researchers and the external partners of the study. The questionnaire was purpose-built by the research team based on extensive previous experience and study in the areas under investigation. This paper is part of a larger study with a larger number of senior researchers and their research centres. The variables for this paper were selected taking into account the objective of the study, the literature and the available variables.

Information in table 1 is somehow redundant with table 2. They might be combined or, table 1 mentioned in supplement material only. The administered questionnaire might be added to supplemental material as well.

Table 1 has been removed and information integrated into the old Table 2.

Minor comments:

page 2 line 48: this affirmation require a reference.

page 2 line 66: A study by... whom?

page 2 line 92: In the study by ... whom?

page 4 line 157-159: unclear. Needs rephrase.

page 4 line 187: SES - needs explanation first time appear in the text

page 4 lines 195-196: unclear. Please rephrase.

page 4 line 202: Paiva et al 2021b. Reference number?

page 5 line 223: CFA. Explanation?

page 5 line 227-229: unclear. Please rephrase and provide details, explanations. Tome et al needs reference number.

page 7 Table 3: what is n.s.? Explanation at the bottom of table.

page 11 lines: 285-287: unclear. please rephrase.

page 13 lines 374-375: unclear. please rephrase.

all comments have been corrected in the document

Reviewer 4 Report

The authors focused on the ecological model explaining the compliance with COVID-19.  The authors identified groups at higher risk; in general, women, older peo-436 ple, and those with lower levels of education have greater difficulty in adapting to the 437 challenges posed by the pandemic.

(1) Note that some references are pre-published on medRxiv and bioRxiv, the authors are suggested to check and update the status of these references.

(2) Are the figures in the paper the author's original drawings or are they copied from existing literature? If the latter, the authors need to mark the citation.

(3) The literature search is weak. The intersection of machine learning, deep learning models, and the related COVID-19 solutions are suggested to be cited and discussed, such as:

[1] A survey on applications of artificial intelligence in fighting against covid-19. ACM Computing Surveys

[2] The impact of COVID-19 restriction measures on loneliness among older adults in Austria. Eur J Public Health

(4) The authors are suggested to provide the data sources of the studies.

(5) In Fig. 1, a brief description of the figure content is suggested in the figure caption.

Author Response

Dear Reviewer
Thank you very much for the comments and suggestions that contribute to the improvement of our work, following the changes made. any clarification or change needed just let us know Best regards Tania

The authors focused on the ecological model explaining the compliance with COVID-19.  The authors identified groups at higher risk; in general, women, older peo-436 ple, and those with lower levels of education have greater difficulty in adapting to the 437 challenges posed by the pandemic.

(1) Note that some references are pre-published on medRxiv and bioRxiv, the authors are suggested to check and update the status of these references.

(2) Are the figures in the paper the author's original drawings or are they copied from existing literature? If the latter, the authors need to mark the citation.

Figure 1 was created by the authors with the aim of illustrating the conceptual ecological model that was analysed and results from this study

(3) The literature search is weak. The intersection of machine learning, deep learning models, and the related COVID-19 solutions are suggested to be cited and discussed, such as:

[1] A survey on applications of artificial intelligence in fighting against covid-19. ACM Computing Surveys

New and more diversified references were integrated

Jianguo, C., Kenli, L., Zhaolei, L., Keqin, L., & Philip Y.. A Survey on Applications of Artificial Intelligence in Fighting Against COVID-19. ACM Comput. Surv. 2022, 54, 8, 158 doi:https://doi.org/10.1145/3465398

Stolz, E., Mayerl, H. & Freidl, W. The impact of COVID-19 restriction measures on loneliness among older adults in Austria. 2021 Eur J Public Health 31(1): 44-49

(4) The authors are suggested to provide the data sources of the studies.

was included extra information in discussion section

(5) In Fig. 1, a brief description of the figure content is suggested in the figure caption.

Figure 1 was created by the authors with the aim of illustrating the conceptual ecological model that was analysed and results from this study

Round 2

Reviewer 2 Report

The authors declined the request to provide the questionnaire based on the excuse that the form is still in the validation phase. This is exactly the reason why the research group needs to disclose the form for peer review and validation. Otherwise, it is impossible to adopt the study design by other research groups. The act of hiding something casts doubts on the scientific integrity of the work.

Author Response

The authors declined the request to provide the questionnaire based on the excuse that the form is still in the validation phase. This is exactly the reason why the research group needs to disclose the form for peer review and validation. Otherwise, it is impossible to adopt the study design by other research groups. The act of hiding something casts doubts on the scientific integrity of the work.

In the attached document there are the questions used to measure the variables used in the article (table 1), we present the Portuguese version and the translated version for reviewers' analysis.

Reviewer 3 Report

The author addressed most of the comments of the first review. I still believe the introduction is too long, and the material and method section could have been better described, but the article may be published in this form. English language and style were significantly improved. 

Conclusions are related to results.

Author Response

The author addressed most of the comments of the first review. I still believe the introduction is too long, and the material and method section could have been better described, but the article may be published in this form. English language and style were significantly improved. 

The introduction has been shortened as indicated in the manuscript. We hope that the method will be further clarified by the questions included in the questionnaire that were at the origin of the variables considered in the manuscript
